# Stay Focused is All You Need for Adversarial Robustness

Bingzhi Chen
Beijing Institute of
Technology, Zhuhai
Zhuhai, China
chenbingzhi.smile@gmail.com

Ruihan Liu
South China Normal
University
Guangzhou, China
liuruihan@m.scnu.edu.cn

Yishu Liu
Harbin Institute of
Technology, Shenzhen
Shenzhen, China
liuyishu@stu.hit.edu.cn

Xiaozhao Fang*
Guangdong University of
Technology
Guangzhou, China
xzhfang168@126.com

Jiahui Pan
South China Normal
University
Guangzhou, China
panjiahui@m.scnu.edu.cn

Guangming Lu*
Harbin Institute of
Technology, Shenzhen
Shenzhen, China
luguangm@hit.edu.cn

Zheng Zhang
Harbin Institute of
Technology, Shenzhen
Shenzhen, China
darrenzz219@gmail.com

## Abstract

Due to the inherent vulnerability of neural networks, adversarial attacks present formidable challenges to the robustness and reliability of deep learning models. In contrast to traditional adversarial training (AT) methods that prioritize semantic distillation and purification, our work pioneers a novel discovery attributing the insufficient adversarial robustness of models to the challenges of *spatial attention shift* and *channel activation disarray*. To mitigate these issues, we propose a robust spatial-aligned and channel-adapted learning paradigm, which we term the "**StayFocused**", that integrates spatial alignment and channel adaptation to enhance the focus region against adversarial attacks by adaptively recalibrating the spatial attention and channel responses. Specifically, the proposed StayFocused mainly benefits from two flexible mechanisms, i.e., Spatial-aligned Hypersphere Constraint (SHC) and Channel-adapted Prompting Calibration (CPC). Specifically, SHC aims to enhance intra-class compactness and inter-class separation between adversarial and natural samples by measuring the angular margins and distribution distance within the hypersphere space. Inspired by the top-$K$ candidate prompts from the clean sample, CPC is designed to dynamically recalibrate channel-wise feature responses by explicitly modeling interdependencies between channels. To comprehensively learn feature representations, the StayFocused framework can be easily extended with additional branches in a multi-head training manner, further enhancing the model's robustness and adaptability. Extensive experiments on multiple benchmark datasets consistently demonstrate the effectiveness and superiority of our StayFocused over state-of-the-art baselines.

## CCS Concepts

• **Computing methodologies → Computer vision problems**.

*Corresponding authors: Guangming Lu and Xiaozhao Fang.

## Keywords

StayFocused; Adversarial attacks; Adversarial training; Adversarial robustness; Spatial alignment; Channel adaptation

**ACM Reference Format:**
Bingzhi Chen, Ruihan Liu, Yishu Liu, Xiaozhao Fang, Jiahui Pan, Guangming Lu, and Zheng Zhang. 2024. Stay Focused is All You Need for Adversarial Robustness. In *Proceedings of the 32nd ACM International Conference on Multimedia (MM '24), October 28–November 1, 2024, Melbourne, VIC, Australia.* ACM, New York, NY, USA, 10 pages. https://doi.org/10.1145/3664647.3681676

## 1 INTRODUCTION

In the past decade, Deep Neural Networks (DNNs) have emerged as indispensable tools for addressing complex real-world challenges across diverse domains within the multimedia field. These applications encompass a broad spectrum of scenarios, ranging from biomedical imaging [3, 4] and face recognition [34] to image retrieval [25]. However, due to the inherent susceptibility of neural networks, adversarial attacks and perturbations [31, 47] can expose significant security vulnerabilities in DNN-based models. Consequently, adversarial robustness has become a crucial metric for evaluating the reliability and trustworthiness of these models.

To enhance the robustness of deep learning models, early studies have concentrated on refining variations against adversarial attacks, such as integrating additional regularization terms [17], introducing core set-based training strategies [8], and adjusting the perturbation size of training data [42]. By employing a large number of adversarial samples as augmented data, adversarial training (AT) approaches [26, 27] have garnered significant attention from the research community. Inevitably, a significant gap exists between the training robustness and test robustness of adversarially trained models [43]. With the advancement of knowledge distillation techniques, adversarial distillation (AD) [16, 32, 46] is dedicated to enhancing the robustness of lightweight networks by distilling valuable insights from adversarially pre-trained models. Despite the progress achieved by pioneering efforts, most mainstream models still suffer from inadequate adversarial robustness.

While semantic distillation and purification have been integral in refining the semantic robustness of adversarially trained models, our research is driven by a fresh cognitive viewpoint beyond semantic interpretation. We demonstrate that the challenges of **spatial attention shift** and **channel activation disarray** posed

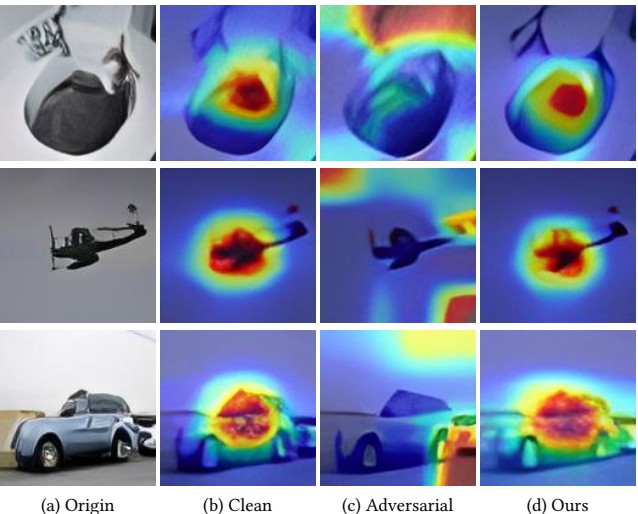

(a) Origin      (b) Clean      (c) Adversarial      (d) Ours

**Figure 1: Visualization of activation maps generated by (a) Original image, (b) Clean sample with non-robust model, (c) Adversarial sample with non-robust model, and (d) Adversarial sample with our work. Compared with conventional models, our StayFocused method effectively directs attention to the most significant regions against adversarial attacks.**

by adversarial attacks and perturbations can be regarded as the main culprits responsible for reducing the robustness of models. As illustrated in Figure 1(b), spatial attention shift refers to the phenomenon where adversarial perturbations cause a displacement in the model's attention towards irrelevant features, significantly impacting its performance and reliability. Intuitively, as shown in Figure 2(a), channel activation disarray arises when adversarial attacks disrupt the activation patterns of individual channels, which can hinder the model's ability to extract meaningful representations from the data. By identifying these challenges, our work strives to provide a new solution to the challenges posed by adversarial attacks, improving the robustness of deep learning systems.

In this paper, we propose a robust spatial-aligned and channel-adapted learning paradigm called "**StayFocused**", which aims to stay focused on discriminative features against adversarial attacks by adaptively recalibrating the spatial attention and channel responses. Specifically, the proposed StayFocused incorporates two flexible mechanisms, i.e., Spatial-aligned Hypersphere Constraint (SHC) and Channel-adapted Prompting Calibration (CPC). *On the one hand*, the main purpose of SHC is to implicitly perform spatial alignment by facilitating intra-class compactness and inter-class separation between adversarial and natural samples. Based on the angular margin measurement within the hypersphere space, it aims to minimize the hyperspherical distribution distance within the same category while maximizing the distinct margin from different categories. *On the other hand*, CPC empowers the network to recalibrate channel-wise feature responses by explicitly capturing the interdependencies among different channels. Benefiting from the top-$K$ candidate class prompts, it dynamically adjusts the channel magnitudes to prioritize important features while suppressing noise and irrelevant features, leading to more accurate representations

against adversarial attacks. By incorporating diverse adversarial objectives related to masking ratio, a flexible multi-head training strategy is also proposed to learn more comprehensive feature representations. Our main contributions are summarized as follows:

- Our work pioneers a novel perspective by identifying the challenges of spatial attention shift and channel activation disarray as critical factors contributing to the insufficient adversarial robustness of deep learning models.
- Two well-designed mechanisms, i.e., SHC and CPC, are proposed to effectively recalibrate spatial attention and channel responses. Additionally, the StayFocused framework can seamlessly incorporate additional branches, enhancing its performance via a multi-head training strategy.
- Our StayFocused is comprehensively evaluated on multiple large-scale datasets, and the promising performance on both clean data and adversarial samples collectively demonstrates its effectiveness over state-of-the-art algorithms.

## 2 RELATED WORKS

### 2.1 Adversarial Attack

In the research community, adversarial attacks can be broadly classified into white-box and black-box attacks. It is noted that our work is dedicated to defending against a variety of white-box attacks [2, 11, 26, 31] while maintaining discriminative capability on clean samples. Sezgedy *et al.* [31] introduced the concept of adversarial samples, which involves adding imperceptible noise to natural samples to cause misclassification by DNNs [33]. After that, numerous influential techniques for white-box attacks have emerged to combat adversarial examples. For example, Goodfellow *et al.* [11] introduced the Fast Gradient Sign Method (FGSM) that utilizes gradient information to identify the most aggressive perturbation within a specified range. Inspired by FGSM, Madry *et al.* [26] proposed a multi-step perturbation strategy called projected gradient descent (PGD) to generate stronger adversarial samples. Different from the previous gradient search perturbation, Carlini Wagner *et al.*[2] proposed an optimization-based attack method as C&W, which is widely used to evaluate the robustness of deep learning models. Furthermore, Croce *et al.* [7] explored an extension of PGD attacks and integrated them with existing attacks, namely AutoAttack, to evaluate the robustness of adversarially trained model.

### 2.2 Adversarial Defense

**Adversarial Training.** Among the various existing defense strategies, adversarial training (AT) [19, 26, 35, 45] is widely acknowledged and employed to enhance model robustness against adversarial attacks and perturbations. Given the classification task based on the batch training set $\mathcal{X} = \{(x_1, y_1), ..., (x_n, y_n)\}$, where $n$ denotes the batch size, each sample $x$ with the ground-truth label $y$ is drawn from the data distribution $\mathcal{D}$. Theoretically, the function of AT can be defined as a min-max optimization problem [9, 10, 36, 49],

$$\min_{\theta} \max_{x' \in \mathcal{B}_\epsilon(x)} \mathcal{L}\left(\mathcal{F}\left(x', \theta\right), y\right), \tag{1}$$

where $\mathcal{F}$ represents a DNN-based model with weight parameters $\theta$, $x'$ is the adversarial example within the $L_p$-norm ball $\mathcal{B}_\epsilon(x) = \{x' :$

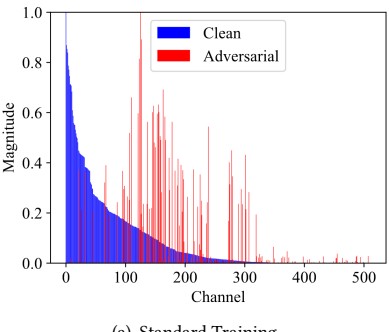 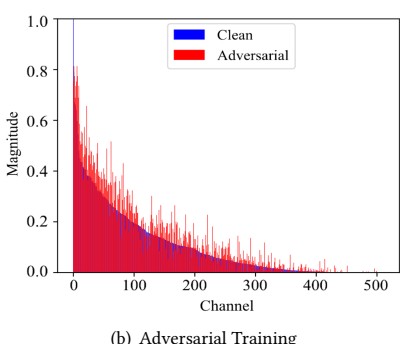 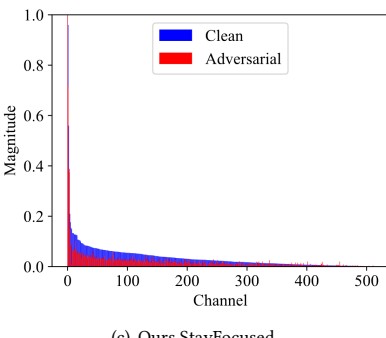

(a) Standard Training                    (b) Adversarial Training                    (c) Ours StayFocused

**Figure 2: Comparisons of the averaged channel magnitudes between standard training, adversarial training, and our proposed method for both natural and adversarial samples. In each plot, the 512 channels are sorted in descending order of magnitude.**

$\|x' - x\|_p \le \epsilon\}$ centered at $x$ and $y$ is its corresponding label. In the context of a classification task, $\mathcal{L}$ refers to the loss function, e.g., the cross-entropy loss. Here, the inner maximization problem depends on adversarial examples $x'$ generated within the $\epsilon$-ball. In contrast, the outer minimization problem optimizes model parameters under worst-case perturbations based on the inner maximization process.

**Adversarial Distillation.** The goal of knowledge distillation is to transfer the knowledge learned by the large robust model (teacher) to the lightweight target model (student), enabling the student model to achieve similar performance with reduced computational resources [12]. Given a well-trained fixed teacher network $\mathcal{T}$ with higher capacity, previous AD works [6, 10, 18, 20, 50] have attempted to incorporate knowledge distillation with AT to enhance the adversarial robustness of the trainable student networks $\mathcal{S}$, which can be formulated as the following optimization:

$$\min_s (1 - \alpha)\mathcal{L}_{\text{CE}}\left(\mathcal{F}_s\left(x\right), y\right) + \alpha\tau^2 \mathcal{L}_{\text{KL}}(\mathcal{F}_s(x'), \mathcal{F}_t(x)), \quad (2)$$

where $\alpha$ is the trade-off factor and $\tau$ is a temperature constant, $\mathcal{L}_{\text{CE}}$ represents the cross-entropy (CE) loss that encourages the student $\mathcal{S}$ to maximize the natural accuracy, and $\mathcal{L}_{\text{KL}}$ denotes the Kullback-Leibler (KL) divergence that aims to minimize the distribution difference across teacher-student domains.

**Advanced Adversarial Robustness.** Recently, variants of advanced defense strategies have been proposed to enhance adversarial robustness. For instance, the channel-wise activation suppressing (CAS) strategy [1] suppressed redundant activations caused by adversarial perturbations. To achieve a better trade-off, Zhang *et al.*[45] decomposed the adversarial prediction error into the natural error and boundary error, proposing TRADES to simultaneously control both terms. Similarly, the Channel-wise Importance-based Feature Selection (CIFS) [41] generated non-negative multipliers for channels to manipulate channel activations for specific layers. In contrast, MART [35] additionally considered misclassified examples during adversarial training. Kim *et al.* [21] proposed a recalibration strategy called Feature Separation and Recalibration (FSR) to recapture its potential discriminative clues. Furthermore, Yin *et al.* [43] proposed an effective method named AGAIN to obtain the attribution span of the model under real and random labels, aiming to enlarge the learned attribution span.

## 3 METHODOLOGY

Technically, the proposed StayFocused is driven by a robustness paradigm shift, specifically focusing on the challenges of spatial attention shift and channel activation disarray posed by adversarial attacks. Figure 3 provides a detailed pipeline of our proposed StayFocused framework, comprising three essential modules: 1) Multi-branches Feature Embedding, 2) Spatial-aligned Hypersphere Constraint, and 3) Channel-adapted Prompting Calibration.

### 3.1 Multi-branches Feature Embedding

To proficiently capture distributed feature representations from input images, a well-established multi-branch architecture is designed as the backbone, comprising adversarial and clean branches. Each encoder is initialized with a pretrained ResNet-18 model. In the clean branches, StayFocused incorporates the concept of mask-denoising [13, 39, 40] to explore the semantic-relevant local image. By robustly capturing localized patterns from randomly masked patches of images, we aim to capture the semantic relevance between different local images through two branches that share parameters, which can be formulated as follows:

$$\min_\phi \mathbb{E}_{(x,y)\sim\mathcal{D}}\ \mathcal{L}_{\text{KL}}\big(f_\phi(x), f_\phi(\text{Mask}(x))\big), \quad (3)$$

where $\phi$ represents the network parameters of the clean encoder.

### 3.2 Spatial-aligned Hypersphere Constraint

Inspired by contrastive learning [5, 14, 29], our SHC mechanism aims to implicitly perform spatial alignment by enhancing intra-class compactness and inter-class separation between adversarial and natural samples. By measuring the angular margins and distribution distance, it is engineered to anchor robust features within the distribution of natural features, while dispersing features from different classes across the hypersphere.

**Typical Contrastive Learning.** Contrastive learning serves as a self-supervised learning paradigm, where the primary objective is to drive the model to draw similar data points nearer within the embedding space, while simultaneously pushing apart dissimilar data points. As mentioned in [5], each instance $x$ contains a set of positive views and a set of negative views [5]. In particular, the

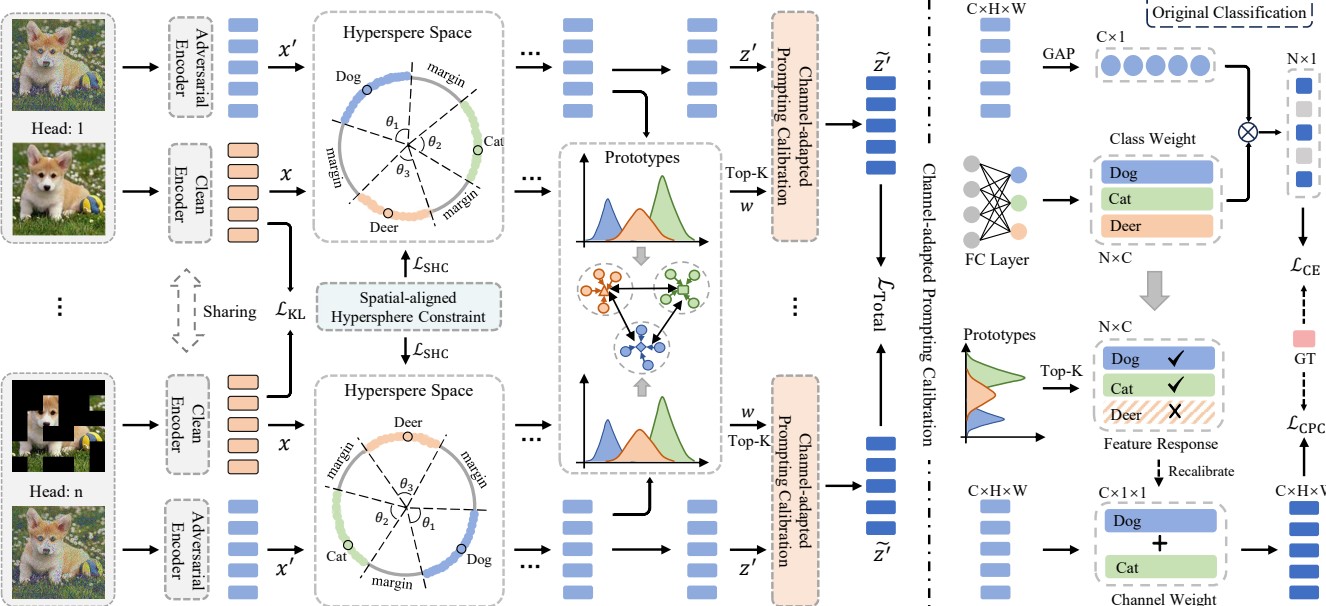

**Figure 3: Illustration of the proposed spatial-aligned and channel-adapted learning paradigm (StayFocused) for combating adversarial attacks and perturbations.** *Left: Multi-branch Feature Embedding* is built on a multi-branch architecture to capture both adversarial and natural feature representations. *Middle: Spatial-aligned Hypersphere Constraint* is designed to facilitate intra-class compactness and inter-class separation between adversarial and natural samples. *Right: Channel-adapted Prompting Calibration* aims to recalibrate channel-wise feature responses by modeling the interdependencies among channels.

contrastive loss function of a positive pair $(x_i, x_j)$ is defined as:

$$\mathcal{L}_{\text{CL}} = -\log \frac{\exp(\text{sim}(x_i, x_j)/\tau)}{\exp(\text{sim}(x_i, x_j)/\tau) + \sum_{k \in \mathcal{N}(i)} \exp(\text{sim}(x_i, x_k)/\tau)}, \quad (4)$$

where $\text{sim}(x_i, x_j)$ denotes a similarity metric between samples $x_i$ and $x_j$, $\mathcal{N}(i)$ represents the set of negative embeddings.

**Definition of Angular Margin.** For the binary classification task, let $\Delta(,)$ represent the angle between each pair of feature embeddings. Suppose the learned adversarial feature $x'$ is given, where $\Delta(x', x^a)$ and $\Delta(x', x^b)$ denote the angles between the adversarial sample and natural samples from different ground-truth categories,

$$\Delta(x^a, x^b) = \Delta(x', x^a) + \Delta(x', x^b), \quad (5)$$

To classify $x'$ in the spherical space, it is necessary to ensure that $\Delta(x', x^a)$ is greater than the angles of the other class,

$$\cos(m \cdot \Delta(x', x^a)) > \cos(\Delta(x', x^b)). \quad (6)$$

As shown in Figure 4, the decision boundary can be formulated as:

$$m \cdot \Delta(x', x^a) = \Delta(x', x^b). \quad (7)$$

where $m \geq 1$ is an integer coefficient. According to Eq. (5) and Eq. (7), the hyper-spherical angular margin $\mathcal{M}$ [23] between classes $a$ and $b$ can be calculated as:

$$\mathcal{M} = |\Delta(x', x^a) - \Delta(x', x^b)| = \frac{m-1}{m+1} \cdot \Delta(x^a, x^b) \quad (8)$$

**Hypersphere Contrastive Learning.** Based on the aforementioned analysis, the essence of the angular margin lies in constraining the arc length on the unit circle, which in turn amplifies the

discriminative power of the features learned on the hypersphere. Geometrically, we leverage the concept that the dot product of two vectors can yield an angle, thereby transforming the optimization of the feature vector into angle optimization on the unit sphere. Formally, the multiplication of the adversarial and nature vectors can be expressed as:

$$x \cdot x' = \|x\|^\top \|x'\| \cdot \cos(\Delta(x, x')). \quad (9)$$

In this way, we can transform the comparison of traditional feature similarity matrices into an optimization problem on a hypersphere with an angular margin $\mathcal{M}$. Compared with standard contrastive learning, the objective function of hypersphere contrastive learning is formulated as follows:

$$\mathcal{L}_{\text{SHC}} = -\log \frac{\exp(\cos(m \cdot \Delta(x'_i, x_i))/\tau)}{\exp(\cos(m \cdot \Delta(x'_i, x_i))/\tau) + \sum_{j \neq i} \exp(\cos(\Delta(x'_i, x_j))/\tau)}. \quad (10)$$

### 3.3 Channel-adapted Prompting Calibration

To combat challenges arising from channel activation disarray, the CPC mechanism is meticulously designed to suppress redundant channels while maintaining the activation of relevant channels. Benefiting from the top-$K$ candidate class prompts, this strategic modulation enables the model to recalibrate its visual attention towards crucial channels, enhancing its resilience and robustness against adversarial attacks and perturbations.

**Top-K Class Prompts.** In parallel with advancing the adversarial branch, we introduce a standard training branch to pinpoint the centroids of distinct class clusters by incorporating category

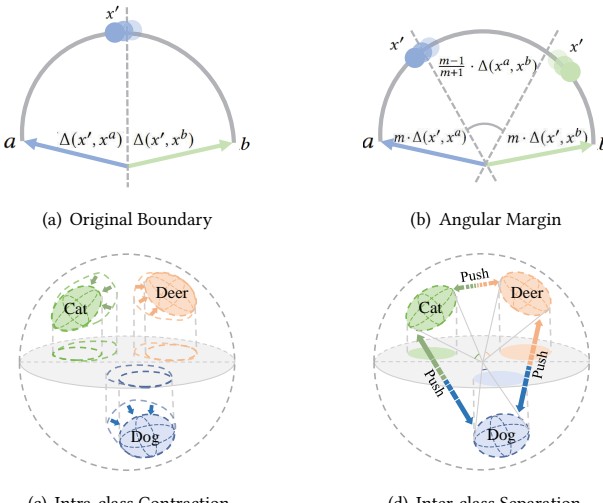

(a) Original Boundary      (b) Angular Margin

(c) Intra-class Contraction      (d) Inter-class Separation

**Figure 4: Details of spatial-aligned hypersphere constraint.**

prototypes during the training phase. It can provide a more detailed representation and understanding of each class's inherent characteristics and distributions in the feature space. After that, we compute the cosine similarity between its feature representation and the prototypes of all classes. As a result, the categories with top-$K$ confidence scores are selected as the class prompts. This selection process ensures that the model focuses on the most relevant and informative classes for each sample, enhancing its discriminative capability and robustness against adversarial attacks.

**Channel-wise Recalibration.** In this part, we initially acquire the preliminary channel feature response $z'$ for each adversarial embedding via a forward inference process. Following the concept of class activation mapping [48], we leverage the top-$K$ class prompts to recalibrate the channel-wise feature responses,

$$\text{Training}: \ \tilde{z}'_l = \alpha \cdot z'_l w^y_l + (1-\alpha)/K \cdot \sum z'_l w^k_l, \quad (11)$$

where $k \in \{1, ..., K\}$, $\alpha$ is a trade-off factor to balance the weights between different classes, and $w^y_l$ and $w^k_l$ denote the channel weight of the $l$-th feature map corresponding to ground-truth class $y$ and top-$K$ candidate classes, respectively. Given that label information and nature samples are unavailable during the testing phase, the class obtaining the highest score in similarity calculation with the prototype is considered as the initial predicted class,

$$\text{Inference}: \ \tilde{z}'_l = z'_l w^{pre}_l, \quad (12)$$

where $w^{pre}_l$ denotes the channel weights of the $l$-th feature map provided by its corresponding class prototype vector. Notably, the internal maximization of the optimization objective depends on estimating the gradients of input pixels based on the loss function. During the adversarial sample generation phase, our CPC strategy is maintained in a frozen state to ensure effective gradient computation for various attack algorithms, without discarding any informative channels.

**Theoretical and Robust Analysis.** Based on Eq. (11), the objective function for classification during the adversarial training

phase can be formulated as follows:

$$\begin{aligned} \mathcal{L}_{\text{CPC}} &= \mathcal{L}_{\text{CE}}(\sigma(\tilde{z}'), y) \\ &= \mathcal{L}_{\text{CE}}(\sigma(\alpha \cdot z'w^y + (1-\alpha) \cdot z'w'), y), \end{aligned} \quad (13)$$

where $\sigma$ represents the softmax activation function, $w'$ denotes the class weights corresponding to candidate classes. By incorporating the cross-entropy loss, $\mathcal{L}_{\text{CPC}}$ can be transformed to:

$$\begin{aligned} \mathcal{L}_{\text{CPC}} &= -\log \left\{ \frac{\exp\left[\alpha \cdot z'w^y + (1-\alpha) \cdot z'w'\right]}{\sum_{j=0}^N \exp\left(z'w^j\right)} \right\} \\ &= -\alpha\left(z'w^y - z'w'\right) - z'w' + \log\left[\sum_{j=0}^N \exp\left(z'w^j\right)\right] \quad (14) \\ &= -\alpha\left(z'w^y - z'w'\right) + \log\left[\frac{\sum_{j=0}^N \exp\left(z'w^j\right)}{\exp\left(z'w'\right)}\right], \end{aligned}$$

where $N$ is the number of classification categories. Suppose $h = z'w^y - z'w'$ [43], Eq. (14) can be reformulated as:

$$\begin{aligned} \mathcal{L}_{\text{CPC}} &= -\alpha h + \\ &\quad \log\left\{ \frac{\exp(zw')\sum_{j=0}^N \left[\exp\left(z'w^j - z'w'\right)\right]}{\exp\left(z'w'\right)} \right\} \quad (15) \\ &= -\alpha h + \log\left[\sum_{j \neq y}^{N-1} \exp\left(z'w^j - z'w'\right) + \exp(h)\right]. \end{aligned}$$

Theoretically, under ideal classification conditions,

$$\nabla_h \mathcal{L} = -\alpha + \frac{\exp(h)}{c + \exp(h)} = 0. \quad (16)$$

Consequently, we can infer the following:

$$z'w^y - z'w' = \log\left[\frac{\alpha \cdot \sum_{j \neq y}^{N-1} \exp\left(z'w^j - z'w^k\right)}{1-\alpha}\right]. \quad (17)$$

According to Eq. (17), a reasonable trade-off mechanism with $\alpha$, e.g., $\alpha = 0.5$, allows the model to focus on the activated features associated with the predicted class while also considering the activated features of related classes. Therefore, it is evident that a comprehensive fusion strategy used in Eq. (11) enhances the model's reliability and robustness against adversarial perturbations.

### 3.4 Multi-head Training and Optimization

Intuitively, our flexible multi-head training approach serves as a pivotal enhancement to the model's capabilities. By integrating insights from various heads, StayFocused gains a richer understanding of the data distribution, leading to improved generalization and resilience against adversarial attacks. Meanwhile, each head focuses on capturing different aspects of the data, allowing the model to understand and adapt to various complexities and nuances presented by adversarial inputs.

**Total Objective Function.** Based on the above analyses, the training objective of the proposed StayFocused approach is a combination of multiple loss functions from different modules, i.e.,

$$\mathcal{L}_{\text{Total}} = \sum \text{Multi-head}\left(\mathcal{L}_{\text{SHC}} + \mathcal{L}_{\text{CPC}} + \mathcal{L}_{\text{KL}}\right). \quad (18)$$

**Table 1: Comparisons of clean accuracy (%) and robust accuracy (%) against various adversarial attacks on the CIFAR-10 dataset.**

| CIFAR-10 | | ResNet-18 | | | | | | | WideResNet-34-10 | | | | | | |
|---|---|---|---|---|---|---|---|---|---|---|---|---|---|---|---|
| Method | Ref. | Clean | FGSM | PGD-20 | PGD-50 | PGD-100 | C&W | AA | Clean | FGSM | PGD-20 | PGD-50 | PGD-100 | C&W | AA |
| AT | ICLR'18 | 84.25 | 55.11 | 46.56 | 44.85 | 44.76 | 48.97 | 41.69 | 84.26 | 58.50 | 56.11 | 55.23 | 55.15 | 54.02 | 51.52 |
| TRADES | ICML'19 | 83.64 | 57.39 | 50.67 | 50.38 | 50.20 | 49.56 | 46.81 | 84.92 | 60.06 | 56.05 | 55.93 | 55.82 | 54.91 | 52.95 |
| CAS | ICLR'21 | 86.79 | 55.99 | 51.49 | 51.77 | 51.04 | 53.66 | 44.23 | 85.37 | 58.96 | 57.84 | 57.68 | 57.43 | 58.47 | 53.25 |
| CIFS | ICML'21 | 83.86 | 58.86 | 51.23 | 49.80 | 48.70 | 50.16 | 43.94 | 84.63 | 59.39 | 58.49 | 56.98 | 56.31 | 55.25 | 52.36 |
| FSR | CVPR'23 | 81.46 | 58.07 | 52.47 | 51.62 | 51.02 | 49.44 | 46.41 | 83.83 | 60.59 | 56.89 | 56.29 | 55.63 | 54.96 | 51.89 |
| AGAIN-PGD-AT | CVPR'23 | 87.88 | 56.87 | 54.43 | 53.62 | 53.13 | 55.80 | 49.31 | 87.36 | 59.80 | 60.73 | 60.04 | 59.83 | 61.52 | 53.19 |
| AGAIN-AWP | CVPR'23 | 86.52 | 62.43 | 59.35 | 59.11 | 58.85 | 61.19 | 51.89 | **90.31** | 62.76 | 62.43 | 62.29 | 62.01 | 68.13 | 53.59 |
| StayFocused | Head=2 | 88.08 | 68.08 | 65.45 | 64.94 | 64.62 | 65.26 | 61.13 | 87.99 | 63.34 | 58.19 | 57.28 | 57.24 | 57.05 | 57.79 |
| StayFocused | Head=3 | 89.02 | 74.72 | 74.19 | 73.55 | 73.39 | **73.16** | 62.10 | 88.31 | 68.95 | 63.83 | 63.68 | 63.67 | 61.66 | 58.10 |
| StayFocused | Head=4 | **89.80** | **76.87** | **75.81** | **75.59** | **74.94** | 72.24 | **67.29** | 89.28 | **77.44** | **77.03** | **76.73** | **75.76** | **70.14** | **62.60** |
| Increased ↑ | - | 1.92% | 14.44% | 16.46% | 16.48% | 16.09% | 11.97% | 15.40% | - | 14.68% | 14.60% | 14.44% | 13.75% | 2.01% | 9.01% |

**Table 2: Comparisons of clean accuracy (%) and robust accuracy (%) against various adversarial attacks on the SVHN dataset.**

| SVHN | | ResNet-18 | | | | | | | WideResNet-34-10 | | | | | | |
|---|---|---|---|---|---|---|---|---|---|---|---|---|---|---|---|
| Method | Ref. | Clean | FGSM | PGD-20 | PGD-50 | PGD-100 | C&W | AA | Clean | FGSM | PGD-20 | PGD-50 | PGD-100 | C&W | AA |
| AT | ICLR'18 | 91.21 | 55.55 | 42.55 | 39.36 | 37.54 | 40.61 | 45.58 | 91.33 | 61.76 | 55.08 | 53.27 | 52.86 | 51.16 | 47.46 |
| TRADES | ICML'19 | 90.99 | 58.10 | 47.12 | 43.83 | 43.55 | 45.48 | 46.29 | 94.89 | 63.27 | 57.88 | 55.32 | 54.88 | 55.04 | 51.53 |
| CAS | ICLR'21 | 90.39 | 65.24 | 51.98 | 44.39 | 43.75 | 53.53 | 47.40 | 91.85 | 62.46 | 58.35 | 56.05 | 55.63 | 56.18 | 48.84 |
| CIFS | ICML'21 | 93.21 | 66.24 | 52.02 | 48.57 | 47.49 | 50.13 | 46.95 | 94.46 | 63.45 | 59.44 | 57.46 | 57.02 | 56.32 | 51.04 |
| FSR | CVPR'23 | 91.28 | 60.46 | 43.94 | 39.74 | 39.01 | 50.22 | 49.27 | 93.46 | 62.87 | 56.71 | 54.84 | 53.36 | 52.68 | 49.97 |
| AGAIN-PGD-AT | CVPR'23 | 92.69 | 65.32 | 60.54 | 55.63 | 53.25 | 58.22 | 51.04 | 94.02 | 64.24 | 60.35 | 58.48 | 59.86 | 62.03 | 53.23 |
| AGAIN-AWP | CVPR'23 | 91.57 | 66.58 | 63.56 | 58.63 | 57.01 | 61.28 | 53.25 | 93.68 | 64.70 | 62.09 | 61.94 | 61.39 | 64.78 | 53.62 |
| StayFocused | Head=2 | 93.14 | 67.79 | 63.81 | 62.79 | 62.54 | 66.03 | 57.72 | 94.50 | 70.51 | 63.27 | 56.02 | 54.19 | 58.34 | 53.28 |
| StayFocused | Head=3 | 93.40 | **72.52** | 67.58 | 66.77 | 66.57 | 65.99 | **58.99** | 95.06 | 79.44 | 73.75 | 68.21 | 65.58 | 70.47 | 54.76 |
| StayFocused | Head=4 | **93.54** | 69.55 | **68.42** | **67.98** | **67.86** | **68.15** | 57.89 | **95.56** | **80.41** | **77.44** | **72.59** | **68.81** | **75.04** | **56.94** |
| Increased ↑ | - | 0.33% | 5.94% | 4.86% | 9.35% | 10.85% | 6.87% | 5.74% | 0.67% | 15.71% | 15.35% | 10.65% | 7.42% | 10.26% | 3.32% |

Through the joint optimization of these losses, our approach can further improve the robustness and reliability of deep learning models. Our experimental results demonstrate that a straightforward summation of the objective loss, without incorporating weight constraints, yields satisfactory outcomes across various benchmark datasets. The training algorithm of StayFocused is shown in our supplementary materials.

# 4 EXPERIMENTS

## 4.1 Datasets, Baselines, and Metrics

In our experiments, we evaluate the effectiveness of our proposed StayFocused framework on two benchmark datasets, i.e., CIFAR-10 [22] and SVHN [28]. In particular, a wide range of state-of-the-art baselines are introduced, including two AT methods, namely AT[26] and TRADES [45], and five advanced adversarial robustness methods, including CAS [1], CIFS [41], FSR [21], AGAIN [43] and AGAIN with AWP [38]. To make a fair comparison, we utilize natural/clean accuracy on natural test samples and robust accuracy on adversarial test samples as the primary evaluation criteria.

## 4.2 Implementation Details

Following the existing studies, we adopt the ResNet-18 [15] and WideResNet-34-10 [44] as the backbone architecture for the proposed StayFocused method.

**Training Phase.** Following the conventional settings in existing works[21], we utilize the stochastic gradient descent (SGD) optimizer with momentum 0.9, weight decay $5 \times 10^{-4}$, an initial learning rate of 0.1 for CIFAR-10 and 0.01 for SVHN, which is divided by 10 at the 75th and 90th epochs. We apply our method to adversarial training PGD-10 (10-step PGD) with a step size of 2/255 and the perturbation $\epsilon$ in the adversarial attack under $L_\infty$-norm is set to 8/255 for all methods.

**Evaluation Phase.** The robustness of the model is evaluated by measuring the correct accuracy of the model under different adversarial attacks. We choose several adversarial attack methods to attack the trained model, including single-step attack algorithm FGSM [11], multi-step attack algorithm PGD [26] (PGD-10, PGD-20, PGD-50 and PGD-100), C&W [2], and AutoAttack (AA) [7]. The

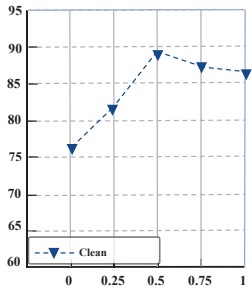 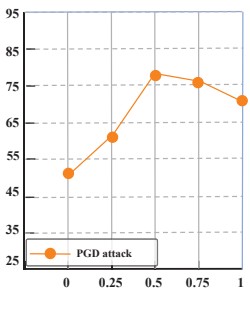

(a) Clean Accuracy      (b) Adversarial Attack

**Figure 5: Ablation study (%) for the proposed StayFocused framework with different $\alpha$ configurations on CIFAR-10.**

maximum perturbation strength of all attack methods under $L_\infty$-norm is set to 8/255 and the step size is $0.1 \times \epsilon$. This comprehensive evaluation ensures a robust assessment of the model's resilience against a wide range of adversarial perturbations.

## 4.3 Comparisons with State-of-The-Art

To evaluate the effectiveness of our StayFocused in enhancing adversarial robustness, we conduct comprehensive experiments against a variety of state-of-the-art baselines on CIFAR-10 and SVHN datasets. The comparative results are summarized in Table 1 and Table 2.

**Evaluation on CIFAR-10 Dataset.** We can observe that our proposed StayFocused method clearly outperforms all the comparative baselines on benchmark datasets. Compared to the current best-performing method, i.e., AGAIN-AWP, StayFocused trained by ResNet-18 model on the CIFAR-10 achieves an average improvement of 3.28% (89.80% vs. 86.52%) in clean accuracy and an average improvement of 16.09% (74.94% vs. 58.85%) under the standard 100-step PGD attack. Particularly, when facing the challenging scenario of comprehensive attack method AA, the classification accuracy of StayFocused shows a 15.4% improvement (67.29% vs. 51.89%) and a 9.01% improvement (62.60% vs. 53.59%) in ResNet-18 and WideResNet-34 on CIFAR-10, respectively. Furthermore, the evaluation results on the ResNet-18 model reveal a substantial improvement of 11.05% in test accuracy (72.24% vs. 61.19%) for StayFocused compared to the AGAIN-AWP baseline under C&W attack. Therefore, the above experimental results show that our StayFocused method effectively uses prior knowledge from the standard training branch to recalibrate visual attention and perform spatial alignment, and has better robust performance compared with SOTA in the same field.

**Evaluation on SVHN Dataset.** To evaluate the generalizability of StayFocused on large datasets, we conduct extensive experiments on the real-world dataset SVHN. The proposed StayFocused framework surpasses the current state-of-the-art approaches by a large margin. In particular, StayFocused achieves substantial improvements in terms of Top-1 accuracy on clean samples, surpassing the state-of-the-art approaches by 1.97% (93.54% vs. 91.57%) and 1.88% (95.56% vs. 93.68%). Consistently, StayFocused increases PGD-100 attack accuracy by 10.85% on ResNet-18 and 7.42% on WideResNet-34. Furthermore, the AA accuracy increased significantly by 5.74% (58.99% vs. 53.25%) and 3.32% (56.94% vs. 53.62%) on the ResNet-18 and WideResNet-34 models respectively, further confirms the

technical superiority of StayFocused in real-world scenarios. These experimental results provide compelling evidence regarding the efficacy and robustness of StayFocused across a range of real-world datasets exhibiting attacks of varying intensity.

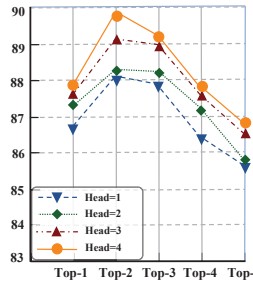 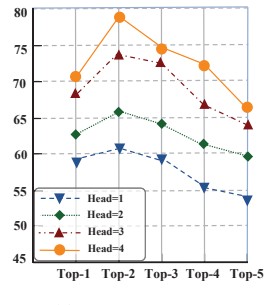

(a) Clean Accuracy      (b) Adversarial Attack

**Figure 6: Ablation study (%) for the proposed StayFocused framework with different top-$K$ configurations on CIFAR-10.**

## 4.4 Parameter Analysis

We study an exhaustive parameter analysis of our proposed Stay-Focused method under different parameter configurations. Specifically, we focus on analyzing the effects of two key parameters: the trade-off factor of $\alpha$ and the hyperparameter of $K$ in Eq. (11).

**Hyperparameter $\alpha$.** The hyperparameter $\alpha$ controls the ratio between true labels and the top-$K$ candidate prompts. As $\alpha$ increases, the model allocates greater attention to feature maps associated with the true label. When $\alpha = 1$, only feature maps relevant to the true label are amplified. Conversely, as $\alpha$ decreases, the model prioritizes feature maps of the top-$K$ candidate prompts, with $\alpha = 0$ exclusively enhancing these maps. We experimented with different $\alpha$ values using ResNet-18 on CIFAR-10 to find the optimal $\alpha$. Through experimentation and analysis in Figure 5. When $\alpha$ equals 0 or 1, the model exclusively attends to the feature map under top-$K$ candidate prompts and those under true labels, respectively. Based on experimental results, $\alpha = 0.5$ yields the optimal outcome. Thus, we adopt $\alpha = 0.5$ during experiments to prioritize the feature map associated with true labels while enabling the model to learn additional feature knowledge.

**Hyperparameter $K$.** The hyperparameter top-$K$ dictates the number of candidate prompts considered. When $K = 1$, only real labels are utilized. As $K$ increases, more pertinent candidate feature maps are incorporated. We vary $K$ and conduct experiments on the CIFAR-10 dataset using ResNet-18 to ascertain the optimal top-$K$. The experimental results are depicted in Figure 6. Observing the experimental results, the best performance is achieved when $K=2$. However, with further increments in the number of top-$K$, the model absorbs an excessive number of redundant feature maps, leading to a reduction in discriminative capability.

## 4.5 Ablation Studies

**Effect on Each Component.** We systematically evaluate the impact of each component on the model's performance. The comparative results are presented in Table 3. "StayFocused w/o CPC" shows reduced adversarial robustness, emphasizing CPC's critical role in suppressing redundant channels and recalibrating visual attention.

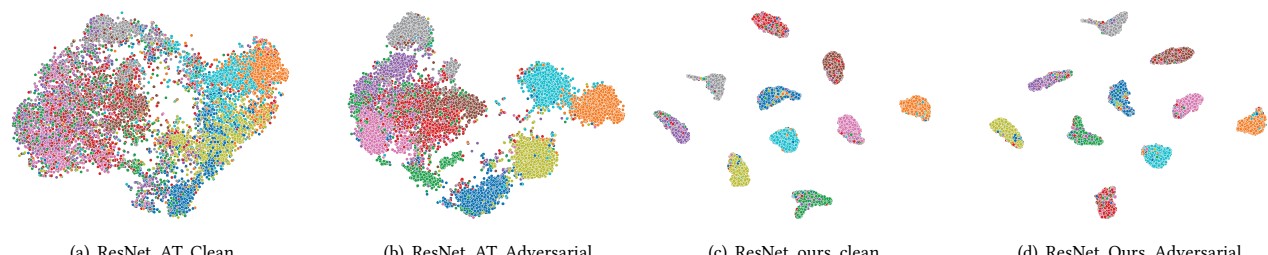

(a) ResNet, AT, Clean    (b) ResNet, AT, Adversarial    (c) ResNet, ours, clean    (d) ResNet, Ours, Adversarial

**Figure 7: T-SNE Visualization of the discriminative features learned by PGD-AT and our StayFocused method on CIFAR-10.**

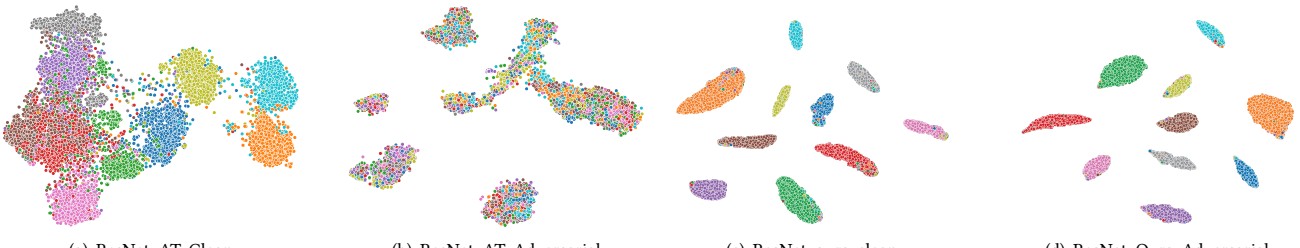

(a) ResNet, AT, Clean    (b) ResNet, AT, Adversarial    (c) ResNet, ours, clean    (d) ResNet, Ours, Adversarial

**Figure 8: T-SNE Visualization of the discriminative features learned by PGD-AT and our StayFocused method on SVHN.**

**Table 3: Ablation studies (%) for the proposed StayFocused framework on the CIFAR-10 dataset.**

| Method | SHC | CPC | Clean | FGSM | PGD-100 | C&W | AA |
|---|---|---|---|---|---|---|---|
| Standard AT | ✗ | ✗ | 84.25 | 55.11 | 44.76 | 48.97 | 41.69 |
| w/o CPC | ✓ | ✗ | 86.75 | 56.29 | 50.61 | 50.40 | 45.43 |
| w/o SHC | ✗ | ✓ | 85.61 | 66.83 | 63.52 | 61.75 | 52.37 |
| StayFocused | ✓ | ✓ | **88.08** | **68.08** | **64.62** | **65.26** | **61.13** |

**Table 4: Comparison of our StayFocused with other contrastive measurement methods. The SHA module is replaced by Cossim [37] and InfoNCE [14].**

| Method | Cossim | vs. | InfoNCE | vs. | SHA |
|---|---|---|---|---|---|
| Clean | 85.86 | | 85.66 | | 88.08 |
| FGSM | 57.45 | | 58.47 | | 68.08 |
| PGD-100 | 52.51 | | 55.79 | | 64.62 |
| C&W | 58.49 | | 63.88 | | 65.26 |

Similarly, excluding the SHC mechanism significantly decreases clean accuracy, as evidenced by "StayFocused w/o SHC".

**Effect on Hypersphere Constraint.** We also comprehensively compare the hyper-sphere contrastive learning used in our SHC mechanism. The comparative results are presented in Table 4. Compared with two popular measurement methods, cosine similarity [37] and InfoNCE [14, 24], experimental results show that SHC substantially improves clean accuracy and adversarial robustness under various adversarial attacks. The results from the ablation studies above underscore the significance of our SHC mechanism in enhancing both intra-class compactness and inter-class separation.

## 4.6 Visualization Results

**Spatial Alignment.** As illustrated in Figure 1, we adopt the class activation mapping approach to identify the relevant attentional visual regions. In contrast to traditional methods susceptible to adversarial attacks, as demonstrated in Figure 1 our approach consistently directs its attention towards the most relevant regions associated with the ground-truth labels.

**Channel Recalibration.** Meanwhile, Figure 2(c) visualizes the averaged channel magnitudes derived from StayFocused. This visualization vividly showcases the efficacy of our approach in counteracting channel activation disarray. By recalibrating the channel-wise feature responses, our method ensures a more coherent and stable representation of visual information.

**T-SNE Visualization.** To further showcase the effectiveness of StayFocused, we employ the T-SNE technique [30] to represent the distributions of feature representations for both clean and adversarial samples. As depicted in Figure 7 and Figure 8, the T-SNE [30] plots resulting from our StayFocused method exhibit a substantial increase in intra-class compactness, while revealing a notable improvement in the separation between different classes.

## 5 CONCLUSION

In this paper, we discovered the phenomenon of adversarial perturbation causing spatial attention shift and channel activation disarray and proposed a StayFocused paradigm to recalibrate the spatial attention and channel responses from a new perspective to solve the above problems. The proposed StayFocused method implicitly conducts spatial alignment and leverages top-K candidate class prompts to maintain focus on discriminative features, combined with a multi-head training strategy to enhance resilience against adversarial attacks. Comprehensive experiments prove that our method is sufficiently effective and universal, and can be integrated into existing advanced frameworks.

## Acknowledgments

This work was supported in part by the National Natural Science Foundation of China (Grant Nos. 62302172, 62176077, and 62176065), in part by the Guangdong International Science and Technology Cooperation Project (Grant No. 2023A0505050108), in part by the Shenzhen Key Technical Project (Grant Nos. JSGG20220831092805009, JSGG20201103153802006), and in part by the Opening Project of GuangDong Province Key Laboratory of Information Security Technology (Grant No. 2023B1212060026).

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
