# OpenReview forum: "Stay Focused is All You Need for Adversarial Robustness"
_acmmm.org/ACMMM/2024/Conference — MM2024 Poster_

### Official Review · Reviewer_HbXg · 2024-05-12

**Rating:** 3
**Confidence:** 3

**Summary:**

This paper raises an interesting finding based on the activation maps to show that the adversarial samples will destroy the current attention regions of the classifier, thus leading to misclassification. Then, the author indicates two root causes: 1) spatial attention shift and 2) channel activation disarray. To solve these problems, the author proposes "StayFocused", an adversarial training framework, to enhance the attention regions of the classifier. The overall experimental results show that it performs better.

However, current experiments cannot show its efficiency due to the weaknesses (detailed in the Sec. Limitation). Meanwhile, there is some confusion about the methodology (also see in Sec. Limitation)). Therefore, we rate it as "borderline reject".

**Strengths:**

1. This paper proposes an interesting finding that the adversarial samples will destroy the correct attention regions of the classifier, thus leading to misclassification. Meanwhile, this paper gives a detailed analysis to show the root causes for this finding, which is innovative.

2. The overall method performs better than the baselines mentioned in this paper.

**Limitations:**

1. The experiments are insufficient. 1) The experiments lack the necessary comparison with SOTA methods. According to the records in the RobustBench for adversarial training (AT), the SOTA performance for AT in CIFAR-10 with the backbone of WideResNet 28-10 against autoattack is 92.44% and 67.31% for standard and robust accuracy, respectively [2]. It seems that this paper's performance does not achieve SOTA. 2) The dataset lacks the CIFAR-100 and ImageNet. To prove the efficiency of AT, it is necessary to report the related experiments on CIFAR-100 and the sub-set of ImageNet.

[1] RobustBench: a standardized adversarial robustness benchmark. Croce, Francesco and Andriushchenko, Maksym and Sehwag, Vikash and Debenedetti, Edoardo and Flammarion, Nicolas. arXiv preprint arXiv:2010.09670, 2022.

[2] Better Diffusion Models Further Improve Adversarial Training. Zekai Wang, Tianyu Pang, Chao Du, Min Lin, Weiwei Liu, Shuicheng Yan. ICML 2023, 2023.

2. There is some confusion in the methodology parts:

-- How do we make sure Eq. 5 is valid on multi-label classification conditions? As mentioned by the author, they assume that for " binary classification", the angle between two categories could be represented by assuming that the adversarial samples are located in the tangent surface between two categories. This is reasonable, but under multi-label conditions, this assumption is strong. It may be better to give more details to illustrate its validity.

-- How do we calculate the scale factor $m$ during training? If I have missed anything, please correct me. I have checked all the details, including the appendix, and cannot find the details about $m$.

-- For $\alpha$ in Eq. 11, the overall theory to prove $\alpha = 0.5$ is little unreasonable. To begin with, according to Eq. 17, let $\alpha =0.5$ cancel it, which seems that it is not necessary to introduce this factor to balance the weights between $w^{y}$ and the top-k candidates in Eq. 11. In this view, the following question is why $w^{y}$ is equally important compared to the top-k candidate? since top-k candidates are not always positive based on the ablation study. Then, what $z_{l}$ means? Based on Figure 3, I guess $z_{l}$ are the feature embeddings from different layers. Eq. 16 will be questioned in this view since this assumption is too strong. As shown in Eq. 11 and Eq. 12, inputs differ during the training and inference phases. The assumption may work in the training phase, but how do we ensure it works in the inference phase? Besides, how can each layer's class weight be calculated during the inference phase?

3. The top-k strategy is a little invalid. The reason to raise this confusion is that the ablation study shows a negative influence when $k > 2$ without further analysis. As the author said, " We leverage the top-k class prompts to recalibrate the channel-wise feature responses." The top-k strategy could be regarded as offering more negative samples to enhance the positive ones. In this condition, more negative samples should increase the overall performance. It may be better to offer more details about this.

**Suitability:**

2

---

### Official Review · Reviewer_PKbZ · 2024-05-23

**Rating:** 4
**Confidence:** 4

**Summary:**

The paper proposes a novel framework called StayFocused for enhancing the adversarial robustness of deep learning models.    The framework incorporates spatial alignment and channel adaptation to improve the focus region against adversarial attacks.    It introduces two mechanisms, Spatial-aligned Hypersphere Constraint (SHC) and Channel-adapted Prompting Calibration (CPC), to recalibrate spatial attention and channel responses.

**Strengths:**

(1) The paper introduces a novel perspective on the challenges of spatial attention shift and channel activation disarray in adversarial attacks, and proposes a unique solution to address these challenges.
(2) The proposed StayFocused framework incorporates innovative mechanisms, SHC and CPC, to enhance adversarial robustness by recalibrating spatial attention and channel responses.

**Limitations:**

The two mechanisms, Spatial-aligned Hypersphere Constraint (SHC) and Channel-adapted Prompting Calibration (CPC) proposed in this paper are a little lack of novelty,  such as [1-2], they have already done the research of adversarial robustness on the hypersphere space based angular margin. If you could clarify the novelty of the mechanisms, i would change my score.


[1] Pang T, Yang X, Dong Y, et al. Boosting adversarial training with hypersphere embedding[J]. Advances in Neural Information Processing Systems, 2020, 33: 7779-7792.
[2] Fakorede O, Nirala A, Atsague M, et al. Improving adversarial robustness with hypersphere embedding and angular-based regularizations[C]//ICASSP 2023-2023 IEEE International Conference on Acoustics, Speech and Signal Processing (ICASSP). IEEE, 2023: 1-5.

**Suitability:**

3

---

### Official Review · Reviewer_UENX · 2024-05-24

**Rating:** 4
**Confidence:** 3

**Summary:**

This paper proposed a spatial-aligned and channel-adapted learning strategy to improve model robustness with two modules: spatial-aligned hypersphere constraint and channel-adapted prompting calibration. Experiments on two datasets are conducted to demonstrate the performance.

**Strengths:**

1. The paper is well-motivated and well-written.
2. The experimental results outperform comparison methods by a large margin.

**Limitations:**

1. More explanations are needed for Section 3.2. How is $\mathcal{M}$ used in the loss of SHC module? How is $\cos(m\cdot\Delta(x’,x))$ computed in Eq. 10 as only $\cos(\Delta(x’,x))$ is available from the dot product.
2. The effect of CPC and SHC is evaluated in the ablation study. What about the ablation study for the multi-head component?
3. The experiments are conducted on two datasets. Are there any results from datasets with larger images?

**Suitability:**

3

---

### Official Review · Reviewer_ajwZ · 2024-05-25

**Rating:** 4
**Confidence:** 2

**Summary:**

This paper proposes a novel paradigm called StayFocused for better adversarial robustness, which consists of a Spatial-aligned Hypersphere Constraint (SHC) and Channel-adapted Prompting Calibration (CPC). Extensive experiments show the efficacy of StayFocused to help models gain better inter-class compactness and intra-class separation. StayFocused achieves significant improvement in robustness against many adversarial attacks.

**Strengths:**

1. The paradigm introduced in the paper is novel and intuitive. The theoretical analysis provided in the paper aligns with the main claim, as well as the experimental results.
2. Experiments are conducted on two datasets and various attacks, showing certain effectiveness of StayFocused. Visualization is also properly employed as an auxiliary for understanding and showcasing the effectiveness of StayFocused.
3. The overall writing of the paper is clear and coherent. The methods and motivations are delineated in a reading-friendly way. Analysis and discussion are also sufficiently thorough and well-covered.

**Limitations:**

1. Experiments are only conducted on relatively small-scale datasets, larger datasets such as ImageNet (or at least Tiny-ImageNet), are absent for validation.

2. Some attacks including adaptive attacks[1] are not included in evaluations. Black-box and transfer-based attacks should also be considered in the evaluation of adversarial robustness.

3. Would the inclusion of MLP in CPC be regarded as a modification to models and thus be unfair for evaluations of other methods?

4. Line 412  should be "smaller than the angles of other classes". Besides, according to Eq.5-7, should Eq.8 be |\Delta(x',x^b) - \Delta(x',x^a)|?

5. In Eq. 15, why didn't the two exp(zw') terms get cancelled out?

6. This paper is that this paper seems to focus explicitly on adversarial robustness in CV, which does not involve any other modality. It would be more suitable if this work provides attempts or justification for expanding to models that involve multi-/cross-modality. could the authors provide the feasibility of expanding their methods to cross-modality?

[1] Tramer, F., Carlini, N., Brendel, W., & Madry, A. (2020). On adaptive attacks to adversarial example defenses. Advances in neural information processing systems, 33, 1633-1645.

**Suitability:**

2

---

### Meta-Review · Area_Chair_uijX · 2024-07-02

**Recommendation:** Accept (Poster)
**Confidence:** 4

**Metareview:**

Overall it is a good paper present an interesting study into adversarial robustness. Some initial concerns of the reviewers were adequately addressed by the rebuttal.

Reviewer PKbZ has a questionable final rating: he/she initially praised the novelty and gave a "weak accept", but then complained about the novelty and changed to "weak reject" without giving a convincing argument. Hence I am discounting this reviewer's rating.